# Geochemical and Statistical Analyses of Trace Elements in Lake Sediments from Qaidam Basin, Qinghai-Tibet Plateau: Distribution Characteristics and Source Apportionment

**DOI:** 10.3390/ijerph19042341

**Published:** 2022-02-18

**Authors:** Haifang He, Haicheng Wei, Yong Wang, Lingqing Wang, Zhanjie Qin, Qingkuan Li, Fashou Shan, Qishun Fan, Yongsheng Du

**Affiliations:** 1Key Laboratory of Comprehensive and Highly Efficient Utilization of Salt Lake Resources, Qinghai Institute of Salt Lakes, Chinese Academy of Sciences, Xining 810008, China; hehaifang19@mails.ucas.ac.cn (H.H.); qinzhanjie@isl.ac.cn (Z.Q.); liqingkuan@isl.ac.cn (Q.L.); shanfsh@isl.ac.cn (F.S.); qsfan@isl.ac.cn (Q.F.); dys@isl.ac.cn (Y.D.); 2Qinghai Provincial Key Laboratory of Geology and Environment of Salt Lakes, Xining 810008, China; 3University of Chinese Academy of Sciences, Beijing 100049, China; 4Institute of Geographical Sciences and Natural Resources Research, Chinese Academy of Sciences, Beijing 100101, China; wangy@igsnrr.ac.cn

**Keywords:** trace elements, sediments, self-organizing maps, positive matrix factorization, Qaidam Basin

## Abstract

The safety of lake ecosystems on the Qinghai-Tibet Plateau (QTP) has attracted increasing attention, owing to its unique location and ecological vulnerability. Previous studies have shown that the aquatic systems on the QTP have been polluted to varying degrees by trace elements. However, little is known of the distribution and sources of trace elements in lakes in the northeast QTP. Here, 57 sediment samples were collected from six lakes (Dasugan Lake, Xiaoqaidam Lake, Kreuk Lake, Toson Lake, Gahai Lake and Xiligou Lake) in the Qaidam Basin, northeast QTP, and the trace elements (V, Cr, Ni, Cu, Zn, As, Ba, Tl, Cd, Pb, and U) were analyzed. The results indicated that Ba, Zn, V, and Cr had a higher content and a wider distribution relative to the other tested elements. Correlation coefficient matrix results showed that the trace elements in the study area were strongly correlated, revealing their source of similarity. Self-organizing maps (SOM, an artificial neural network algorithm) results indicated that the degree of pollution in Xiaoqaidam Lake was the highest, and that of Dasugan Lake was the lowest. Furthermore, all sampling points were clustered into four categories through K-means clustering. The positive matrix factorization (PMF) results indicated that atmospheric deposition and anthropogenic inputs were the main trace elements sources in these lakes, followed by traffic emission and geological sources. Collectively, trace elements of six lakes in Qinghai-Tibet Plateau presented high-content and pollution characteristics. This research provides a scientific basis for better water environment management and ecological protection on the QTP.

## 1. Introduction

The Qinghai-Tibet Plateau (QTP), known as the Third Pole of the Earth, is the largest geographical unit on Earth with an altitude of above 4000 m. The environment and ecosystem in QTP are sensitive and fragile, and have been increasingly affected by variations in global or regional climate [1,2]. Meanwhile, intensive human activities including regional urbanization and industrialization, agricultural development, animal husbandry, and tourism have also profoundly affected the eco-environment in QTP and the related biogeochemical cycle of trace elements [3,4,5]. QTP harbors numerous lakes, which cover an area of about 46,500 km^2^ (accounting for 1.8 % of the QTP area) [6]. The lake-rich QTP has significant environmental impacts on regional and global hydrological cycles through water vapor exchange.

Lakes are important natural resources and play key functions in climate regulation, and providing a site for tourism in QTP [7]. With rapid industrialization and urbanization, lake water pollution has become a serious, worldwide environmental problem [8,9]. In the ecological environments of lakes, sediment serves as a “storage bank” for trace elements, and directly reflects the degree of their contamination as well as the migration, transformation and enrichment of trace elements across ecosystems. Sediments are sensitive indictors, not only as carriers of elements but also of potential secondary contamination in aquatic systems, as evidenced by a substantial body of literature [10,11,12,13,14,15]. Trace elements are mainly concentrated in suspended solids and other media, which become an indicator that reflects the spatial and temporal changes in water pollutants [16]. The process of trace elements releasing from the sediments into the water, and the transfer from water back to sediments, is a bidirectional, reversible process [17]. While the transfer from water to sediments is the dominant aspect of this process, environmental conditions such as changes in salt concentration, redox conditions and changes in pH cause trace elements to be released from suspended solids and/or sediments and cause secondary pollution [18,19]. Trace element pollutants deposited in sediments have the most direct and toxic effects on sediments and organisms in water bodies. In addition, due to mobilization and bioaccumulation of trace elements, they can pass upwards to humans through the food chain, thereby further affecting their health [20,21].

Previous studies have focused on trace elements in different countries and drainage basins, mainly focusing on their concentration and distribution characteristics, ecological risk assessment, source apportionment, chemical morphology and bioavailability, and remediation and treatment. Regarding trace elements found in sediment, researchers have done a lot of work according to three basic evaluation methodologies: Chemical, biological and comprehensive [22,23,24]. Collectively, these approaches have been used to determine the source, spatial variation, distribution characteristics, risk evaluation, and other characteristics of trace elements [25,26]. The source of trace elements remains complex, and their distribution has strong spatial heterogeneity. Given this, we used a Positive Matrix Factorization (PMF) method to identify the main pollution factors and judge the relevant sources according to previous studies. We then sought to analyze the internal relationship between the elements and sampling points through self-organizing maps (SOM) and K-means clustering.

The Qaidam Basin is located on the Qinghai-Tibet Plateau and covers an area of 221,000 km^2^. It is an important, protected area for China and the world, owing to its fragile ecological environment. The Qaidam Basin is the most extensively disturbed area in the entire QTP. The basin is rich in mineral resources such as salt lakes, energy, and metals, and is also an area rich in rare resources [27]. Moreover, the basin is also located at the center of the Qinghai-Tibet Railway, which has many crisscrossing provincial highways and other roadways. However, systematic studies regarding the distribution and contamination characteristics of trace elements pollution in lake sediments in the Qaidam Basin remain scarce. Given this, we selected six plateau lakes in the Qaidam Basin as the research areas. The objectives of this study were two-fold: (1) To clarify the degree and distribution of trace element pollution in the sediments of the six identified lakes in the Qaidam Basin, located in the Qinghai-Tibet Plateau; (2) To determine the correlation between different trace elements and deduce the artificial sources and/or natural sources of any identified trace elements in sampled lake sediments. Taken together, the results presented here will provide further scientific support for more research into extremely sensitive, geographic areas like the Qinghai-Tibet Plateau.

## 2. Materials and methods

### 2.1. Study Area

The Qaidam Basin is located in the northeastern Qinghai-Tibet Plateau and is the highest and one of the largest intermountain basins in inland China (Figure 1). This area is subject to a Plateau continental climate, with high evaporation and low precipitation. Regarding the latter, precipitation is mainly concentrated in the months of June, July and August; overall, dust activities are pervasive [28]. The basin is surrounded by tall mountains, with the Riyue Mountains in the east, the Qilian Mountains in the north, the Kunlun Mountains in the south, and the Altun Mountains in the northwest. Collectively, these mountains form an irregular centripetal catchment basin. The basin is affected by topography and neotectonic movement and has developed water systems, with a total of more than 30 rivers and 33 lakes, including salt lakes [29]. This study focused on six lakes as the study area—Dasugan Lake, Xiaoqaidam Lake, Kreuk Lake, Toson Lake, Gahai Lake, and Xiligou Lake—all located in the northern corner of the Qaidam Basin. Of the six lakes, Kreuk Lake and Xiligou Lake were the only freshwater lakes, with the remaining identified as saltwater.

### 2.2. Sampling and Measurements

In this study, 57 sediment samples were sampled from the six previously identified lakes using a grab sampler. Sediment with a thickness of 4–10 cm on the surface layer was obtained with a grab sampler, sealed in a polyethylene bag, and stored in a refrigerator back to the laboratory for treatment. Among these, Dasugan Lake is a large salt lake with an area of 108 km^2^, including the Dasugan Lake Nature Reserve. A total of 10 samples (D_1_–D_10_, 98°26′42.08″–98°27′47.19″ E, 36°49′17.05″–36°51′23.41″ N) were collected from the north bank of the lake to the center of the lake. The water depth ranged from 2.2 to 4.6 m, with an average water depth of 3.7 m. Xiaoqaidam Lake covers an area of 71.5 km^2^ and is a salt lake with coexistence of solid and liquid phases. Ten sediment samples (X_1_–X_10_, 95°26′46.20″–95°28′39.37″ E, 37°28′16.30″–37°28′47.54″ N) were collected from the west bank of the lake. The water depth ranged from 4 to 4.5 m, with an average water depth of 4.45 m. Kreuk Lake covers an area of 57.4 km^2^ and is a fresh water lake. Seven samples (K_1_–K_7_, 96°54′08.36″–96°54′35.71″ E, 37°17′09.00″–37°18′49.19″ N) were collected from the north bank to the center of the lake. The water depth ranged from 2.3–4.5 m, with an average depth of 3.52 m. Toson Lake covers an area of 18.76 km^2^ and is a typical inland saltwater lake. A total of 10 samples (T_1_–T_10_, 96°50′30.57″–96°53′37.98″ E, 37°04′06.96″–37°06′48.78″ N) were collected from the southwest corner of the lake to the center of the lake. The water depth ranged from 2 to 23 m, with an average water depth of 11.6 m. Gahai Lake is a salt lake with an area of 37 km^2^. Nine samples (G_1_–G_9_, 97°31′07.12″–97°33′11.03″ E, 37°08′05.36″–37°08′20.78″ N) were collected from the west bank to the center of the lake. The water depth ranged from 3–19 m, with an average depth of 11 m. Xiligou Lake is a freshwater lake with an area of 33.15 km^2^. Eleven samples (L_1_–L_11_, 98°26′42.08″–98°27′47.19″ E, 36°49′17.05″–36°51′23.41″ N) were from the south bank to the north bank. The water depth ranged from 0.7–3.3 m, with an average depth of 1.96 m (Appendix A). These six lakes are distributed from west to east on the northern Qaidam Basin.

During the pre-treatment process, the weeds and stones in the sediment were removed with the resulting sediment then freeze-dried using a freeze dryer. Then, the sediment was ground with an agate mortar. Finally, the samples were individually passed through a 100-mesh nylon sieve. In the laboratory and before any measurements, the digestion tube and other instruments were soaked in pure nitric acid and 20% nitric acid solution for more than 24 h. Then approximately 0.3 g of sediment was microwave-digested with a mixture of 8 mL HCl, 2 mL HNO_3_, and 2 mL HF. All acids are pure premium grade. The digestion temperature of the first digestion stage was 150 °C; the second was 190 °C. In the first and second digestion stages, the climb time was 15 min, and the hold time was 10 min.

After digestion, an acid distilling instrument was used to distill the acid at 160 °C. When there was 2–3 mL of liquid remaining, 2 mL HClO_4_ was added; the evaporation treatment was stopped; and the mixture was then cooled to room temperature. Finally, the solution in the digestion tube was transferred to a centrifuge tube. The wall of the tube was rinsed several times with small, constant volumes (50 mL) of ultrapure water. The contents of V, Cr, Ni, Cu, Zn, Ba, Tl, Cd, Pb, and U were analyzed using an inductively coupled plasma mass spectrometry (ICP-MS) and an inductively coupled plasma optical emission spectrometer (ICP-OES). The test accuracy was controlled by blank and parallel samples and controlled against the certified reference samples GSS-3(GBW07403) from the National Research Center (Beijing, China) to guarantee accuracy of the experimental measurements. Each sample was measured three times. Moreover, the measurement accuracy was ±10%, and the measurement error was <5% between the measured and the verified values. Sample processing and analysis methods were performed according to the Soil Environmental Quality-Risk control standard for soil contamination of agricultural land (GB 15618-2018) [30].

### 2.3. Statistical Analysis

#### 2.3.1. Self-Organizing Maps (SOM)

The SOM algorithm is an autonomous learning method and an unsupervised learning neuron network model proposed by Kohonen in 1982. As such, it is called the Kohonen algorithm. The entire learning process is conducted in the input sample space and is measured by Euclidian distance [31].

The analysis process is separated in a series of steps, the first being to find a set of center points (also called codebook vectors). Each object of the data set is then mapped to the corresponding center point according to the principle of most similarity. Different from the general neural network training based on the reverse transfer of the loss function, this approach uses a competitive learning strategy to gradually optimize the network by relying on the competition between neurons. The use of a neighborhood function to maintain the topological structure of the input space means that the two-dimensional mapping includes the relative distance between the data points [32]. Adjacent samples in the input space are then mapped to adjacent output neurons, and the unified distance matrix (U-matrix) is usually used as the mapping of the probability density of data points in the input space on a two-dimensional plane when outputting the result [16].

This SOM method realizes an independent analysis of data by the relevant program and classifies the neuron’s similar data into one category, obtaining the classification result. This method greatly eliminates the inaccurate effects caused by human experience and various statistical errors, and is widely used in data mining and sample classification. At present, some studies have shown that the SOM clustering method is suitable for cluster analysis of trace element pollutants and spatial heterosexual analysis [16,33,34].

In this study, the SOM toolbox in MATLAB was used to analyze the content of trace elements in the sediments, and the SPSS software was used to perform k-means clustering on the SOM clustering.

#### 2.3.2. Positive Matrix Factorization (PMF)

PMF is a relatively mature quantitative source apportionment method used for the analysis of pollutant sources in environmental fields. It was first proposed by Paatero in 1994 and was originally used for ambient particulate matter source identification [35]. Recently, it has been widely used in processing environmental data, including 24-h speciated PM2.5, size-resolved aerosol, deposition, air toxins and sediment [36]. Compared with other receptor models, PMF has the following two main advantages: (1) It incorporates the variable uncertainties usually associated with environmental sample measurement; (2) It overcomes the problem that other receptor models have, i.e., principal component analyses are prone to indecipherable factor loads such as non-negative [37].

Given this, the potential sources of the trace elements identified in the six lakes in the Qaidam Basin were analyzed by a PMF 5.0 model. Concretely, the mathematical model matrix is as follows:(1)$$\mathrm{Xij}=\sum_{k=1}^{p}g_{\mathrm{ik}}f_{\mathrm{kj}}+e_{\mathrm{ij}}$$
where X_ij_ is the measured value of the chemical composition of j in the sample i; g_ik_ is the contribution value of the source k to the sample i; f_kj_ is the calculated value of the chemical composition of j in the source k; e_ij_ is the residual error of the chemical composition of j in sample i [36].

The purpose of the PMF model analysis is to minimize Q. Q is defined as:(2)$$Q={\sum_{i=1}^{m}\sum_{i=1}^{n}\left[\frac{X_{\mathrm{ij}}-\sum_{k=1}^{p}g_{\mathrm{ik}}f_{\mathrm{kj}}}{u_{\mathrm{ij}}}\right]}^{2}$$
where Q is the objective function defined by PMF; u_ij_ is the uncertainty of chemical composition j in sample i [36].
(3)$$u_{\mathrm{ij}}=\sqrt{{\left(EF\times c\right)}^{2}+{\left(0.5\times MDL\right)}^{2}}\;(c>\mathrm{MDL})$$
(4)$$u_{\mathrm{ij}}=5/6\times MDL\;(c\;\leq \;\mathrm{MDL})$$
where c is the concentration of trace elements; EF is the error function; MDL is the detection limit of the method [36].

In this research, PMF5.0 software developed by the U.S. Environmental Protection Agency was used to analyze the main sources of trace metals in sediments.

## 3. Results and Discussion

### 3.1. Summary Statistics of Trace Elements Identified in Lake Sediments

The total trace element content in the sediments of the six lakes in the Qaidam Basin are shown in Table 1 and Figure 2. The coefficient of variation (CVs) can be used to compare the degree of dispersion of the data. It is the ratio of the standard deviation of the original data to the mean of the original data. Generally, trace elements exhibited inconspicuous spatial and element-specific variety. The CVs of each element of the six lakes was ranked 7–38%, showing relatively low spatial variation. However, the spatial variability of Ba and U in Dasugan Lake was determined to be moderate at 46% and 51%, respectively.

The trace elements concentrations in sediments obtained from the six lakes sediments in the study area were arranged in intervals as follows: 11–553, 82.7–440, 57.2–234, 39.5–233, 116–571, 125–351, 257–2748, 0.59–4.1, 0.16–1.46, 24.4–166 and 14.8–201 (mg/kg) for V, Cr, Ni, Cu, Zn, As, Ba, Tl, Cd, Pb and U, respectively. Furthermore, the average trace elements content of the six lakes were as follows: V (331), Cr (259), Ni (141), Cu (130), Zn (333), As (212), Ba (1716), Tl (2.32), Cd (0.78), Pb (91.1), and U (52.3) (mg/kg).

In addition, the highest content values of eight of the identified elements—V, Cr, Ni, Cu, Zn, Ba, Tl, and Pb—all appeared in Xiaoqaidam Lake. Comparatively, the lowest content values of eight of the identified elements—V, Ni, Cu, Zn, As, Tl, and Cd—were all found in Dasugan Lake. Given this, we determined that of the six plateau lakes, the pollution level of Xiaoqaidam Lake was relatively high, while that of Dasugan Lake was relatively low. A possible explanation may be that Xiaoqaidam Lake is a famous tourist attraction, located at the intersection of two highways. Moreover, Chaidan Town, where the lake is located, is abundant in mineral resources. In contrast, Dasugan Lake is a nature reserve that is less affected overall by human activities.

As shown in Figure 2, the graphical form of the violin plot with a box was employed as a visual display for the basic statistics of V, Cr, Ni, Cu, Zn, As, Ba, Tl, Cd, Pb and U contents in plateau lake sediment samples. This chart combines the characteristics of a box chart with a density chart. The integral shape of the violin chart with the box represents the estimated density of the data distribution of a given element in a sampled lake.

As shown in Figure 2, the contents of Ba, Zn, V, and Cr in the sampled sediments of the plateau lakes were all significantly higher than those of other elements. Ba content was the highest element in the six lakes’ sediment samples, indicating that Ba was the most commonly occurring element. Most of the lakes in the study area were either sulfate or carbonate-type lakes. For example, the water chemical types of Xiaoqaidam Lake, Gahai Lake, and Xiligou Lake were all sulfate type [38]. Under the influence of a sedimentary environment (such as water depth, redox, biological action, etc.), Ba ions entering the lake often interact with sulfate ions and carbonate ions, and then precipitate to form enrichment [39,40,41], so Ba naturally exists as either a sulfate or carbonate. Judging from the overall shape of the violin plot, the violin plot of each element in the plateau lake was narrow. This indicated that the content data distribution was relatively discrete, without the appearance of any abnormal points.

As shown in Table 1, except for the average values of Cd elements in the six lakes, which were lower than the local background values, the concentrations of other elements in the study area were much higher than their corresponding local background value [42]. These showed that the degree of trace element pollution in the study area was relatively high. For comparison, the Cr, Ni, Cu, Zn, As, Cd and Pb contents of lakes in other regions of China and some representative countries on other continents were shown in Table 1. As shown, the average contents of Cr, Ni, As and Pb in the six lakes in the study area were all higher than those of other additional lakes in different parts of the world, the contents of Cu, Zn and Cd was higher than that of domestic lakes but lower than that of some world lakes [10,18,35,43,44,45]. Collectively, this indicated that the amount of fouling of the lakes in the Qaidam Basin was relatively high and at a high level when compared to other lake systems.

**Table 1 ijerph-19-02341-t001:** Summaries of trace element concentrations (mg/kg) in sediments, background values (mg/kg) in the study area and comparison with other selected water systems around the world (mg/kg).

Lake		V	Cr	Ni	Cu	Zn	As	Cd	Ba	Tl	Pb	U
Dasugan Lake	Mean	185	140	80.6	57.8	153	179	0.25	1127	1.0	37.5	60.9
	Max	306	207	108	73.8	210	235	0.32	2450	1.4	69.0	107
	Min	114	93.1	62.4	39.5	116	126	0.16	661	0.59	24.4	15.7
	SD	59.9	34.4	14.3	11.9	28.5	28.0	0.06	519.6	0.30	14.2	31.0
	CV(%)	32	25	18	21	19	16	24	46	30	38	51
Xiaoqaidam Lake	Mean	489	391	205	204	499	229	0.96	2404	3.4	145	31.9
	Max	553	440	234	233	572	280	1.12	2749	4.1	167	47.1
	Min	335	260	140	132	330	200	0.57	1988	2.6	109	19.8
	SD	71.0	53.4	30.1	32.0	75.8	29.8	0.17	239.1	0.5	17.2	6.7
	CV(%)	15	14	15	16	15	13	17	10	16	12	21
Kreuk Lake	Mean	365	285	148	138	338	201	0.65	1900	2.0	85.9	26.4
	Max	536	380	190	179	421	239	0.82	2346	2.6	107	31.9
	Min	197	139	86.7	83.3	198	159	0.40	359	1.12	52.8	14.8
	SD	119.2	91.9	36.8	35.1	88.2	28.9	0.16	666.4	0.59	22.4	6.5
	CV(%)	33	32	25	25	26	14	24	35	30	26	24
Toson Lake	Mean	242	180	99.9	85.6	295	220	0.84	1491	2.16	85.6	31.9
	Max	310	240	126	106	435	352	1.22	2346	3.12	109	52.6
	Min	130	82.7	57.2	54.0	196	144	0.43	257	1.40	64.3	17.5
	SD	58.1	49.9	21.6	18.1	58.7	54.1	0.24	564.9	0.49	14.1	9.7
	CV(%)	24	28	22	21	20	25	29	38	22	16	30
Gahai Lake	Mean	393	310	170	152	397	215	1.12	1876	3.05	110	32.9
	Max	459	356	197	178	462	312	1.46	2300	3.34	126	42.6
	Min	276	209	105	87.8	273	173	0.52	1630	2.63	74.1	16.6
	SD	62.8	53.7	34.2	32.5	68.55	48.9	0.34	239.1	0.23	16.7	8.3
	CV(%)	16	17	20	21	17	23	30	13	7	15	25
Xiligou Lake	Mean	316	248	145	144	322	231	0.87	1499	2.35	82.7	130
	Max	493	384	204	213	452	280	1.14	2377	3.41	128	202
	Min	151	136	89.2	81.5	183	191	0.57	711	1.35	44.0	48.7
	SD	111.4	86.4	41.8	45.1	94.8	29.6	0.23	513.05	0.70	28.5	45.14
	CV(%)	35	35	29	31	29	13	26	34	30	34	35
Background values ^a^		71.8	70.1	29.6	22.2	80.3	14	1.37	411	0.59	20.9	2.99
Taihu lake, China ^b^		—	87.9	53.9	59.1	140	13.6	1.03	—	—	71.7	—
YR, China ^c^		—	77.2	25.9	46.5	149	25.9	0.42	—	—	37.8	—
Rz and VK, Slovakia ^d^ Kozmalovce, Slovakia (12)		—	62.5	38.9	230.2	490	49.1	2.0	—	—	72.4	—
Reference lake, USA ^e^		—	65.0	31.0	58.0	216	—	0.65	—	—	73.0	—
Qarun lake, Egypt ^f^		—	14.4	55.6	39.1	117	—	1.26	—	—	21.2	—
C Coast, India ^g^		—	110	28.0	76.5	78.7	—	19.8	—	—	49.6	—

Note: SD: Standard Deviation; CV: Coefficients of variation; ^a^ [42]; ^b^ [35]; ^c^ Yangtze River, China, [10]; ^d^ Ruzin and the Velke Kozmalovce, Slovakia, [18]; ^e^ [44]; ^f^ Qarun lake, Egypt, [43]; ^g^ Coromandel Coast, India, [45].

### 3.2. Correlational Analysis between Trace Elements

The correlation between the elements of the selected lakes in the study area is presented in Figure 3. In this chart, the size of the circle represents the significant level of the correlation (larger size, higher significant level), while the coefficients represent the strength of the correlation (higher coefficients, stronger correlations). Regarding the expression of the degree of correlation, it is usually expressed by the range of the absolute value of r: 0–0.19 is a very weak correlation; 0.2–0.39 is weak; 0.4–0.59 is moderate; 0.6–0.79 is strong, and 0.8–1 is very strong.

From the correlation coefficient matrix of the six lakes, most elements had either a strong or a very strong correlation with another element (e.g., V and Cr (r = 0.99, *p* ≤ 0.01), Cr and Zn (r = 0.95, *p* ≤ 0.001), Zn and Pb (r = 0.97, *p* ≤ 0.001), and Ni and Cu (r = 0.98, *p* ≤ 0.001)). These results indicate support for the following three problems. First, these elements coexisted in geochemical processes under different environmental conditions [46]. Simultaneously, they also illustrated that these trace elements may have homology. From the perspective of natural sources, the trace elements found in the Qinghai-Tibet Plateau mostly originated from long-distance transportation of atmospheric deposition and natural weathering. To a certain extent, trace elements in the soil around the lake were transferred to the lake, with internal migration increasing trace element content in the sediment [28,47]. Finally, these correlations may also reflect similar content levels among the various elements [10].

It is worth mentioning that the correlations between As, Ba, U, and other elements were all relatively weak. There was no obvious correlation between As and other elements, which is a similar finding to other research [10,48]. The predominant reason for this finding is that As mainly derives from anthropogenic sources, such as industrial activities [35,49]. As mentioned earlier, Ba naturally exists as either a sulfate or carbonate, different from other elements in the form of the occurrence. This may be the reason for the negligent relationship of Ba with other elements. Finally, U had no obvious enrichment characteristics in the lake sediments and had no correlation with any other elements [50].

### 3.3. Self-Organizing Maps (SOM)

In this study, the SOM algorithm was used as the feature extraction of the content and distribution of trace elements obtained from the study area. The quantization error (QE) and topographic error (TE) values were used to control the accuracy of the model [16]. After running, the QE value and TE values were 0.246 and 0.016, respectively, indicating that the results had good topology retention [51]. Meanwhile, a U-matrix was obtained, and 12 component files of the 44-unit map (4 × 11) were formed (Figure 4). Through the K-means clustering of the SOM results, the data were further refined, and the 57 sampling points were grouped into four categories (Figure 5). In each graph, the hexagon at a specific location corresponds to the same mapping unit. The column labels in Figure 4 and Figure 5 are the same, representing the normalized concentration index.

From the component planes (Figure 4), the similar color distribution and arrangement indicated that V, Cr, Ni, Ca, Zn, Tl, and Pb all had strong correlations, indicating a certain degree of homology between these elements. It is worth noting that the high concentrations of V, Cr, Ni, Cu, Zn, Ba, Tl, and Pb appeared in sampling points such as X_1_-X_9_, suggesting the high-content characteristics of Xiaoqaidam Lake. Comparatively, the low concentrations of V, Cu, Zn, As, Tl, and Cd appeared in sampling points D_1_, D_2_, D_4_, D_6_, D_7_, D_8_, and D_9_, indicating the low-content characteristics of Lake Dasugan.

The internal connection between the sampling points in the study area is shown in Figure 5. Four clusters were determined by the K-means algorithm and are displayed in different colors according to the Euclidean distance. These results indicated that 22 sampling points (e.g., T_5_–T_8_, G_5_–G_7_, D_9_, K_2_, L_4_) belonged to the first category; 14 sampling points (e.g., D_1_–D_6_, L_2_, T_9_) belonged to the second category; X_1_–X_4_, G_4_, T_10_, K_4_, D_8_ and the other 19 elements were classified as the third category; T_4_, K_7_ belonged to the fourth category. These results may have been the result of changes in hydrological and hydrodynamic conditions, as well as gravitational flows caused by the tilt of the lake bottom [52].

### 3.4. Source Apportionment by Positive Matrix Factorization (PMF)

The PMF5.0 model developed by the U.S. Environmental Protection Agency was used to identify possible sources of trace elements obtained from sampled lakes in the Qaidam Basin. Before the model operation, uncertainty calculations were first performed. Since the concentration of each element was greater than the detection limit, the uncertainty calculations used Formula (3). Next, both the trace element concentration and uncertainty files were imported into the model. Simultaneously, the factor of the lake was set to 3, representing three different sources.

After many runs and adjustments, it was found that when the number of iterations was 17 and the number of seeds was 13, the minimum Q value could be determined. The Q (Robust) was close to the Q (true), and the residual error was basically controlled between 0 and 4, indicating that the model had a good fit. Meanwhile, the R^2^ of each trace element was between 0.83–0.99 (except As (0.25)). This indicated that the model had a high accuracy, which further showed that the PMF method was well-suited to the analysis of sediment sources in the sampled plateau lakes (Table 2; Figure 6).

Factor 1 explained only 23.0% of the component with the loading U (99.8), As (39.6) and factor loads of other elements were quite low (Table 2). Uranium and As are trace elements in the soil, mainly derived from the parent material of the soil [4]. In geochemical baseline studies, the natural sources of trace elements in soil and sediments produce baseline concentrations [35]. Low loads indicated that human activities had little contribution to the sources of trace elements in these sediment samples. Given this, factor 1 may be related to geological sources.

Factor 2 accounted for 29.0% of the total contribution and was mainly characterized by the high loading of Cd (58.2), Tl (45.7), Pb (42.6), Zn (37.8), and Cu (36.6) (Table 2). The study area was adjacent to the Qinghai-Tibet Railway and multiple national and provincial highways. Meanwhile, some lakes in the study area are well-known tourist attractions, with tourists and cars constantly moving through the area. Fuel combustion, oil combustion, and brake pad wear will cause enrichment of Cd, Pb, Zn, Cu, and other elements [52]. For example, the emission of Zn in automobile lubricants, the decomposition of trace element components in automobile tires, Cd is mostly derived from engine oils, abrasion of brake linings, and tearing of tires, and the emission of Cd and Cu will all cause enrichment of relevant trace elements [48,53]. Pb mainly comes from the release of leaded gasoline and kerosene [23]. Thus, factor 2 was interpreted as the traffic emissions source.

Factor 3 had the largest contribution and 48.0% consisted of high loading Ba (73.9), As (59.9), Cr (59.7), V (58.9), Ni (52.2), Pb (51.4), and Zn (51.0) (Table 2). Particles containing trace elements diffuse in air and are deposited or transferred into sediment by dust deposition, precipitation, and runoff [4,54]. The study area was located in a transitional zone between aridity and semi-arid conditions. Aeolian sand can be transported to lakes and watersheds through both dry and wet deposition methods. A small number of volatile compounds (e.g., Pb, Cr, Zn, and other trace elements) are closely related to particles. During long-distance transportation, they are mainly deposited along with other particles [23]. Studies have shown that Zn, Pb, Cr, Ni, As, Ba, V, and Pb are known to be associated with traffic [55,56,57]. For example, Ba is often used in diesel engines and other internal combustion engine cleaning agents [58]. Moreover, studies have shown that South Asia has serious air pollution problems, owing to its rapid industrial development. This pollution is then transported to the Qinghai-Tibet Plateau through long-distance monsoon air masses [59,60]. Given this, factor 3 was inferred to be a mixed source of atmospheric deposition and anthropogenic inputs.

The results of this source contribution distribution revealed that the contribution rate of sediment trace element contamination sources caused by man-made activities in this area was comparable to that of natural sources. These results also indicated that along with the improvement of transportation facilities, industrial and mining activities and tourism activities have now entered into this unspoiled, resource-rich primitive environment. As such, human activities have already begun to have an impact on the original ecological environment [1,4,28].

## 4. Conclusions

Trace element contamination in lakes has become an important global environmental issue. However, there is a lack of research on trace element pollution in the lakes of the Qaidam Basin in the Qinghai-Tibet Plateau and four of the six lakes in the study area have not yet been studied. This study determined the pollution factors and pollution levels of trace elements in the plateau lake sediments, the correlation between these trace elements, and the internal connections between the sampling points. These data were also used to conduct a source analysis regarding pollution factors. The results indicated that U, As, Pb, Zn, Ba, Cr, and Cd were the main contributing factors; moreover, that there was a strong correlation between these specific trace elements. Remarkably, atmospheric deposition and anthropogenic input were the main sources of lake trace elements in the study area, followed by the traffic emissions and geological sources. Moreover, the sediment trace element content in the study area was higher than the local soil background values. This indicated that the study area has suffered from contamination, which was similar to the study results of trace element pollution in aquatic systems in other areas of the Qinghai-Tibet Plateau. This may be related to the increasing human activities in the study area. Research on remote lakes will help to understand the overall status of trace element pollution in the lakes of the Qinghai-Tibet Plateau. In future work, more extensive research will be done on trace element pollution in the aquatic systems of the Qinghai-Tibet Plateau. Research in the region will aid in a better, macroscopic understanding of the global pollution landscape. This is of great significance for the overall prevention of trace element pollution in China and elsewhere.

## Figures and Tables

**Figure 1 ijerph-19-02341-f001:**
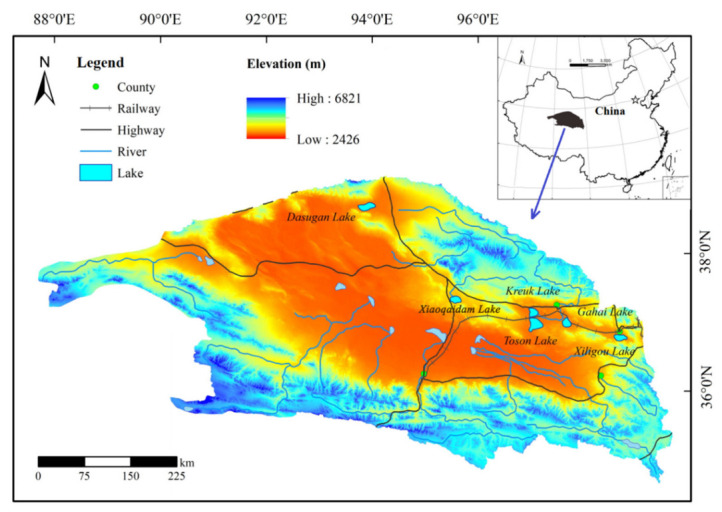
Overview map of the six study lakes in the Qaidam Basin.

**Figure 2 ijerph-19-02341-f002:**
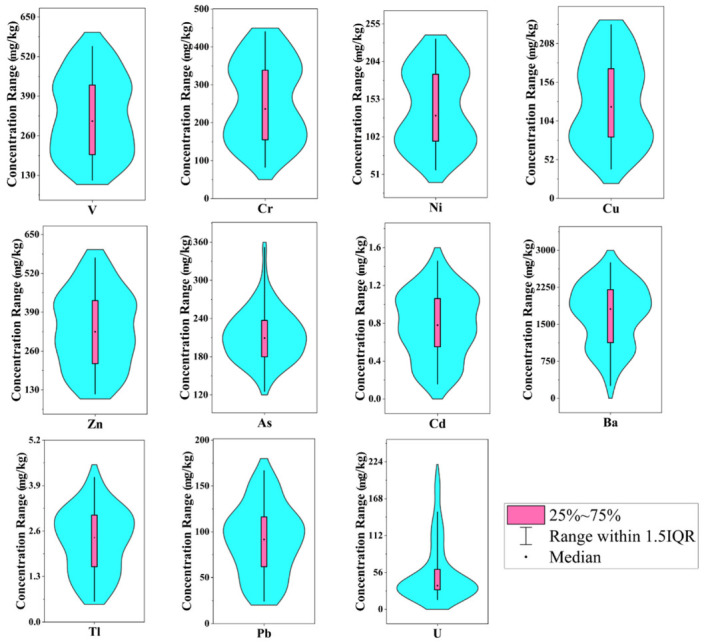
Violin plot with box is used to show the variations and its probability densities of trace elements (mg/kg) in the sediment of the six selected lakes in the Qaidam Basin. Note: The thick black bar in the middle represents the quarter-digit range; a thin black line extending from it indicates that it follows the 1.5IQR rule. White dot indicates the median.

**Figure 3 ijerph-19-02341-f003:**
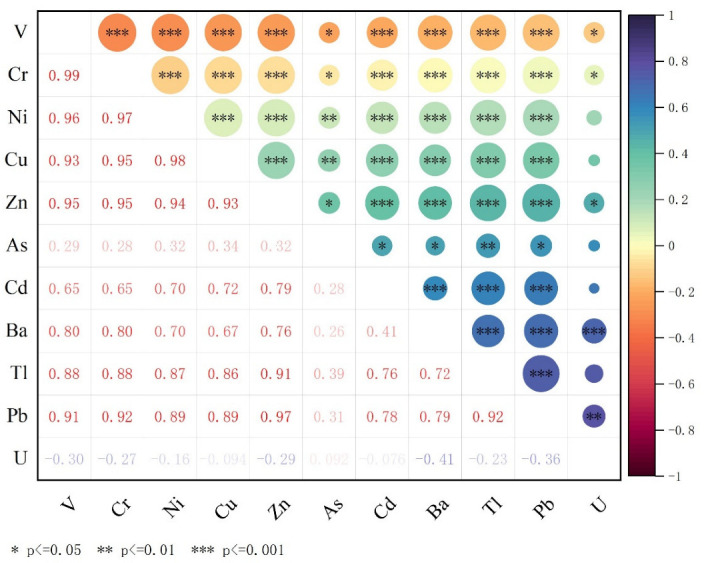
Correlation coefficient matrix between trace elements co-occurrence for sediment in lakes in the Qaidam Basin (* Represents *p*-value ≤ 0.05; ** Represents *p*-value ≤ 0.01; *** Represents *p*-value ≤ 0.001).

**Figure 4 ijerph-19-02341-f004:**
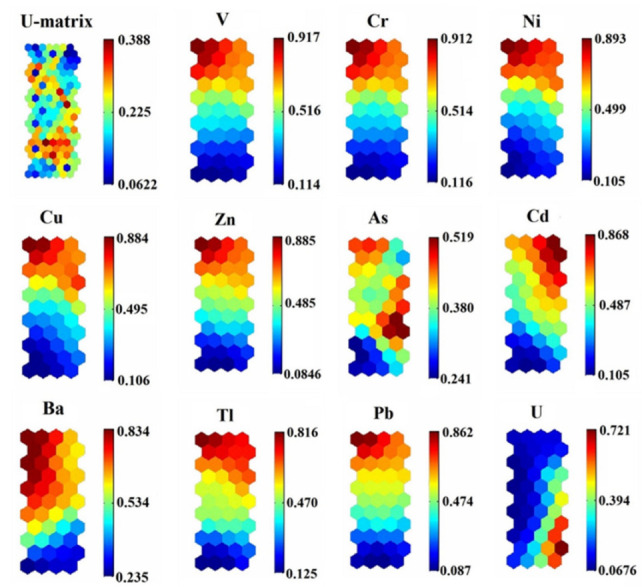
Self-organizing mapping of trace elements. Note: Column labels represent the normalized concentration index. Similar samples are mapped close to each other while different samples are separated. These are represented by different colors. This mapping is based on Euclidean distance.

**Figure 5 ijerph-19-02341-f005:**
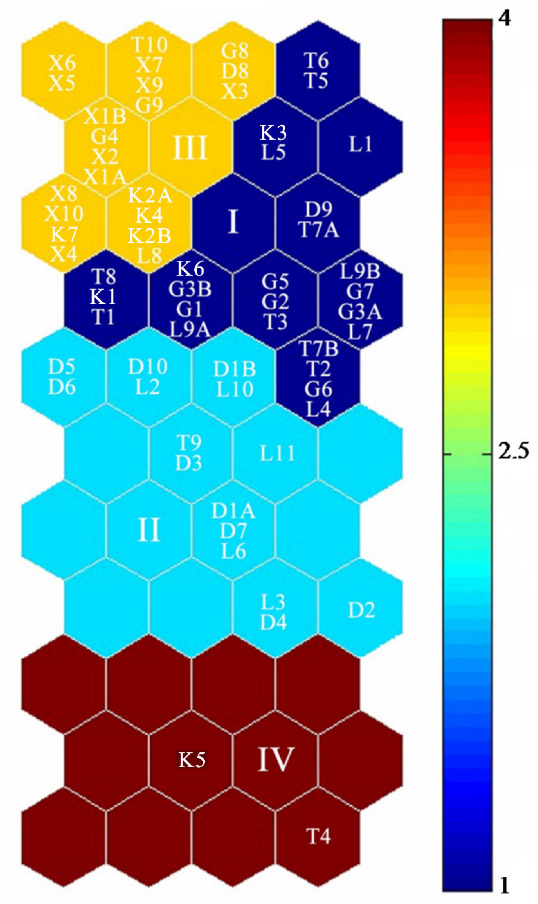
Spatial heterogeneity of trace elements in each sampling point based on SOM. Note: T_1_–T_10_ represents 10 sampling points of Toson Lake; G_1_–G_9_ represents 9 sampling points of Gahai Lake; D_1_–D_10_ represents 10 sampling points of Dasugan Lake; K_1_–K_7_ represents 7 sampling points of Kreuk Lake; X_1_–X_10_ represents 10 sampling points of Xiaoqaidam Lake; L_1_–L_11_ represents 11 sampling points of Xiligou Lake.

**Figure 6 ijerph-19-02341-f006:**
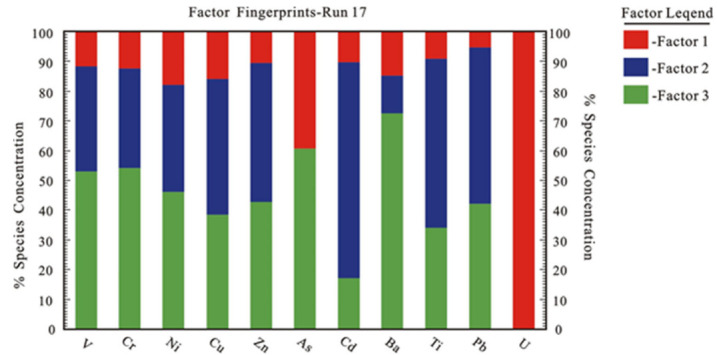
Factor fingerprints of contributions of the three factors identified by PMF.

**Table 2 ijerph-19-02341-t002:** Contribution rate (%) results of each factor and element as calculated by the PMF model.).

Element	Source Contribution Rate/%			
	Factor 1	Factor 2	Factor 3	R^2^
V	12.1	28.9	58.9	0.9567
Cr	12.8	27.4	59.7	0.9671
Ni	18.5	29.3	52.2	0.9473
Cu	16.8	36.6	46.6	0.9316
Zn	11.3	37.8	51.0	0.9724
As	39.0	1.1	59.9	0.2529
Cd	11.8	58.2	30.1	0.8852
Ba	14.7	11.5	73.9	0.8289
Tl	10.1	45.7	44.2	0.8795
Pb	6.1	42.6	51.4	0.9513
U	99.8	0.2	0.0	0.9873
Total contribution rate/%	23.0	29.0	48.0	

## Data Availability

Data is contained within this article.

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
